# Longer Work Shifts, Faster Forward Rotation—More Sleep and More Alert in Aircraft Inspection

**DOI:** 10.3390/ijerph18158105

**Published:** 2021-07-30

**Authors:** Tarja Hakola, Paula Niemelä, Sari Rönnberg, Annina Ropponen

**Affiliations:** 1Finnish Institute of Occupational Health, P.O. Box 40, FI-00032 Työterveyslaitos, Finland; annina.ropponen@ttl.fi; 2Finnair Health Services, P.O. Box 15, 01053 Finnair, Finland; paula.niemela@finnair.com (P.N.); sari.ronnberg@gnhearing.com (S.R.); 3Division of Insurance Medicine, Department of Clinical Neuroscience, Karolinska Institutet, 171 77 Stockholm, Sweden

**Keywords:** working hours, age, vigilance, aviation, stress

## Abstract

The purpose of this intervention study is to compare sleep, alertness, and work ability among aircraft inspectors working under two different shift schedules. The original schedule was forward rotating: MMM – – EEE – NNN – – – (M = morning, E = evening, N = night, – = day off). The new schedule was fast forward rotating: MEN – – with 10-h shifts. The baseline data were collected before the schedule changed, and the follow-up data 12 months (*n* = 10, Group A) or 5 months (*n* = 13, Group B) after the change. Three of subjects were women and average age was 46.6 years (range 31–58). The surveys included questions on sleep quantity, sleep quality, severe sleepiness, alertness, perceived stress, current work ability, and satisfaction with the shift schedule. The results indicated that in the new schedule, the sleeping times were longer and sleep loss was less. Moreover, shift specific severe sleepiness decreased, and alertness during shifts improved. Compared to baseline, perceived stress was lower and work ability was better. Satisfaction with the shift system had also improved. To conclude, the quickly forward rotating shift system might be beneficial in terms of increased sleep length and improved alertness and overall well-being especially among older aircraft inspectors.

## 1. Introduction

Several working hour characteristics in shift work influence the worker’s well-being, safety, and even health [1]. Alertness and sleepiness are related to the time of the day, as well as the duration and timing of the work [2]. For example, sleep may be shortened or disturbed due to early morning shifts, but also in combination with short recovery times between shifts, and during night shift periods [3,4]. In turn, if sleep is insufficient, it may increase acute fatigue and in long-term may also lead to several health problems [1] as well as safety risks [5,6].

Guidelines to arrange shift work are in general level as well as detailed i.e., “few night shifts (maximum of 3)” [7,8,9], but it is not clear how to combinate different work shifts. Shift work arrangements i.e., shift systems can be designed to rotate slower or faster, backward or forward, with longer or shorter work bouts. Some characteristics of shift work, such as number of successive night shifts, short shift intervals, and evening shifts are assumed to be strenuous due to enhanced fatigue and variated sleep problems [10]. In practice the shift schedule is a compromise between work demands and workers’ preferences especially for the number of successive free days. Designing the shift work to be tolerable by employees, needs more knowledge of different shift systems and their effects on human performance in various sectors.

24/7 services at aviation require that the defect control and technical solutions must be available during maintenance hours, mostly by night hours, and during daytime hours for flights operating both domestic and abroad. The key objectives of technical operations are safety, reliable flight operations, and cost-efficient maintenance processes. To ensure the defect control and analysis within the fleet is high-demanding task with time-pressure. As a basis for this study, the smooth operation of the technical inspectors with their tasks was one of the reasons for redesigning the shift schedule. To support the inspector’s well-being at work and balance between work and private life were important topics, too. Furthermore, adjustments for shift schedule were assumed to assist operating hours at nights for airplane maintenance while grounded. To our knowledge the research of schedule change like this in real life situation have hardly been done before.

The purpose of this study was to evaluate the effects of the changes in the shift schedule measured as sleep, alertness, and work ability among aircraft inspections. The original goal of changing the shift schedule was to increase the maintenance time during times of the day, when airplanes do not fly.

## 2. Material and Methods

### 2.1. Study Design

This study was an intervention based on the change of shift schedule with follow-up. The change of shift schedule was bargained as a local agreement with the employer and representatives of inspectors. The trial period was 12 months (Group A), and 5 months (Group B) due to different timetables in local agreements.

The participants of this study, aircraft inspectors are so called trouble-shooters who are responsible for the airworthiness of the fleet of the airline company. They solve all the problems or alarm situations emerging in airplanes during maintenance or even during flights. The total number of inspectors was 31.

The intervention of this study, the change of shift schedule was that the slower rotating shift schedule was changed to faster rotating, and work shifts were lengthened from 8.5 h to 10 h according to the demands of work. The baseline data were collected before the schedule changed and the follow-up data after the change (see details in the Figure 1).

Earlier shift schedule was slow forward rotating: MMM – – EEE – NNN – – (M = morning shifts 5.30–14.00, E = evening shifts 13.30–22.00, N = night shifts 21.30–6.00, – = day off). The new shift schedule was fast forward rotating: MEN – – (M = morning 5.30–15.30, E = evening 12.00–22.00, N = night 20.00–6.00, – = day off). The new shift schedule was a modification of the schedule that has been studied earlier in airline company [11,12,13,14].

### 2.2. Subjects

A total of 23 subjects out of 31 (response rate 77%) participated in both surveys. Three of participants were women and the average age was 46.6 years (range 31–58). Ten subjects were included in the Group A and 13 subjects in the Group B (Table 1). There was a statistically significant group difference between the trial Groups A and B in the mean age and shift work experience.

### 2.3. Questionnaire

The questionnaire was modified from Standard Shiftwork Index [15]. It included general biographical information as age, sex, chrono type, work experience, occupation, and years in shiftwork (Table 1).

Among the dependent variables of this study, we assessed sleep quantity by questions about habitual sleep length as “How many hours do you usually sleep per 24 h in workdays and days off?”, and sleep need as “How many hours of sleep do you need to be alert the next day?”, accuracy of ½ h [16]. Sleep loss was calculated as the mean difference between hours of self-rated need of sleep and sleep length in workdays [17].

The shift-specific retrospective alertness [18,19] was measured by 9-point rating scales (1 extremely alert–9 very sleepy, KSS) [20] as average in every two hours during work shifts. KSS ratings ≥ 7 (%) were defined as severe sleepiness, being critical level for safety due to impaired performance [21].

Current work ability compared with life-time best was measured by single item (scale 0–10, point 0 “cannot currently work at all”, and point 10 “work ability at its best”) of Work Ability Index (WAI) [22]. Shift work satisfaction was measured by a 5-point scale (1 very satisfied–5 very dissatisfied). Perceived stress was measured by a 5-point scale (1 not at all–5 very much) [23].

### 2.4. Statistical Analysis

Statistical analyses were carried out through the analysis of variance with repeated measurements, using the Statistical Analysis System (SAS, ver. 9.4). We used a linear mixed model for repeated measurements containing the baseline and follow-up ratings as the between-subjects factor. Two main effects and their combined effect were tested. The effect of change in shift schedule (intervention) on dependent variables (sleep, alertness, wellbeing) was tested between baseline and follow-up measurements. A significant *p*-value indicates the difference between the initial and final situation, and we also reported F-value for parameter estimates. The effect of two trial groups (Groups A and B, trial periods 12- or 5-months, respectively) indicated the different events between the groups. The combination effect (schedule change*trial group) reveals that the schedule change affects the two groups in different ways.

The group difference for means between the trial Groups A and B was tested with *t*-test for continuous variables. Due to significant differences in age and shift work experience, variable of two age groups (≤45 and ≥46 years) was used as a covariate in the variance analysis. The analyses of variance were made separate for the indicators of sleep, alertness, and wellbeing. The distribution of the variables in this study was mainly normal, but due to small sample size few exemptions existed. However, this violation of normal distribution should minimally affect our results.

A *p*-value of <0.05 indicated a statistically significant result throughout the study. The results show all the main effects obtained in the schedule change and the differences between the groups when they are statistically significant. The combination effects (schedule change*trial group) on all dependent variables were statistically non-significant throughout the study, and no detailed results of interactions are therefore presented.

## 3. Results

In this study the difference between two shift schedules was clear. The fast forward rotating shift schedule was better for sleep and alertness than the earlier, original shift schedule with slower forward rotation.

Estimates of sleep differed significantly between the earlier and new shift schedules (Table 2). Self-estimate of sleep length on working days was, on average, short, about 6 h, but extended to more than 7 h on average in the new schedule. On free days, subjects slept longer than working days, on average 8 and 9 h. The need for sleep was estimated to be the same for both shift schedules, at about 8 h. The difference between the need for sleep and habitual sleep length i.e., sleep loss was nearly 2 h, but it shortened to one hour in the new schedule. 

No statistically significant differences in sleep-related variables were observed between Groups A and B (Table 2), except for the sleep loss that differed significantly between Groups A and B, mean 0.7 h vs. 2.1 h, *p* < 0.04, respectively. The combination effects on sleep variables were insignificant.

The alertness of the subjects differed from shift to shift and due to the timing of the shift, however the alertness varied in the same pattern in both shift schedules (Table 3). Subjects were most alert during evening shifts and sleepier in both morning and night shifts. They were less alert at the beginning of the morning shift, then picked up until at the end of the morning shift the alertness weakened again. In the evening shift, alertness gradually deteriorated toward the end of the shift, while in the night shift, alertness deteriorated sharply during the night shift, being at its worst in the morning hours.

Alertness during shifts was systematically and significantly improved in the new shift schedule. The average alertness during the morning shift was “neither alert nor sleepy” (5.1), the evening shift “alert” (3.4) and the night shift “some signs of sleepiness” (5.9) in the earlier schedule and in the new schedule during the morning shift on average “rather alert” (4.4), evening shift “very alert” (2.5), and the night shift “rather alert” (4.5). Differences in alertness estimates at comparable time points across the shifts were statistically significant (Table 3), indicating the effect of schedule change. Only at the end of the evening shift, at 22.00 o’clock, alertness estimates did not differ significantly between the shift schedules.

The differences between Groups A and B were statistically non-significant at all time points (Table 3), except for the morning shift at 8.00 and 10.00 o’clock, where Group A estimated the alertness to be significantly better than Group B (3.4 vs. 5.2, *p* < 0.02, and 3.1 vs. 5.0, *p* < 0.009, respectively). The combination effects on alertness variables were insignificant.

Severe sleepiness (percentage of KSS ≥ 7) was prominent during both morning and night shifts (Figure 2). More than 40% of subjects rated their alertness beyond the critical level in the morning shift. Toward the end of the night shift, almost everyone (60–90%) assessed their alertness above a critical level. In the new schedule, there were clearly fewer people with poor alert, at most less than 20% in the morning shift and at most 60% at the end of the night shift. In the evening shift, severe sleepiness did not occur in either shift schedule.

The satisfaction with the shift schedule improved significantly (Table 4). On average, the earlier schedule was “fairly dissatisfied” while the new system was “fairly satisfied” (median 4 vs. 2, respectively). Groups A and B did not differ in this respect. Perceived stress was rated on average “somewhat” in the earlier schedule and “only slightly” in the new schedule (Table 4). In experiencing stress, Groups A and B differed significantly from each other (*p* < 0.02), the former rated stress as “little” and the latter as “somewhat” (median 2 vs. 3). Work ability assessments indicated that among the older participants, the work ability increased (mean 7.7 vs. 8.6, Table 4) during new schedule compared to earlier schedule. The work ability assessments of Groups A and B did not differ significantly. The combination effects on wellbeing variables were insignificant.

## 4. Discussion

The aim of this intervention study was to evaluate the effects of changes in the shift schedule (i.e., two different rapid forward rotating shift schedules) on sleep, alertness, and work ability among aircraft inspectors. The trouble-shooters are highly qualified professionals in time pressure and face demands of cognitive performance all day through. The shift schedule was changed by extending all three shifts by 1.5 h during the day, when the morning shift ended later and the evening shift started earlier in the afternoon, while the night shift started earlier in the evening. Thus, the most strenuous shift change-over time between the night shift and the morning shift was intact. Lengthening the work shifts and changing the speed of rotation at same time helped the subjects to sleep more during the work period. Longer sleep also enhanced alertness and perceived work ability of the subjects.

The results are in line with an earlier study [11,13] in which a very quickly forward rotating shift system increased sleep length and improved alertness although the previous shift system was in line with the recommendations. There were some differences in working times compared with the earlier studies of rotation [11,12,13,14] and current intervention: the timings of shifts, the length of shifts, and direction of rotation of basic schedule (backward/forward), mainly due to timetables in air traffic.

As a part of the aircraft inspectors work, the defect control and technical solutions must be available during maintenance hours, mostly by night hours, and flights both domestic and abroad during daytime hours. Hence the timing of the work shifts is dependent on the flight schedules, the maintenance, and control for airworthiness to be done before taking offs. In this study the start of morning shift was exceedingly early in both shift schedules. Due to the early morning start of work, the sleep before the morning shift is reduced, and during the successive of morning shifts the sleep loss cumulates [1,24]. This may be one of the reasons for strenuousness of the earlier, slower rotating schedule. The cumulative sleep loss may also affect performance (i.e., alertness and wellbeing) during the night shift period [25]. During the successive night shifts the cumulative sleep loss is also potential, so vigilance during both morning and night shift bouts is affected. Alertness stays better, when the cumulative sleep loss is prohibited by the shift system with very quickly forward rotation [26].

One of the main innovations of the new shift schedule design was the overlap between the morning and evening, and evening and night shifts, opposite to the normal organization of work shifts having change-over at the same time. During the first 6.5 h of 10-h shifts, the most urgent operations are performed, and during the last three hours there is time to complete the tasks and duties at work avoiding over-time. Furthermore, enough time exist for all the other tasks (reporting, planning, and coordination etc.). The overlap of work shifts also gives flexibility in combining work and private life. Another innovation was the flexible use of the evening shift. In unpredictable situations, like sudden sickness, air traffic problems, etc., the evening shift is possible to swap with morning or night shift, for example changing shift order from MEN to MNN or MMN if necessary.

Due to the increase in the length of the work shifts in the changed schedule, the night shift was extended to cover evening hours, which means that the night shift should end early enough determining the morning shift to start early. This highlights the balance in starting and ending times, which are especially important to be decided locally i.e., at the workplaces to provide possibility to account for local circumstances related to, for example, commuting or avoiding rush-hours etc.

First, the most obvious limitations of the study were the small number of participants and selected individuals, which limit the generalizability of the results. However, most participants had long experience in three-shift work as well as in their occupation, which increases the reliability of the results. The results are consistently positive, so the intervention can be considered a success. This might have to do with the fact that the participation rate in the study was high, so subjects were well represented in their occupational group. Furthermore, we applied the follow-up design in which both in the baseline and follow-up situations the same participants were measured.

One of the limitations is related to the study design which was not optimal due to variation in the duration of the intervention, i.e., Group B had short follow-up. The reason for the discrepancy in the follow-up time is the fact that the Group B was originally a control group but wanted to move to a new rotation system because the positive experiences of Group A were obvious. However, as usual, carrying out the study in real working life is challenging and requires compromises, and the change of the shift schedule in Group B could not be denied before the end of the one-year experiment.

Groups A and B differed in two respects, one was the duration of the intervention, and the other was the age of the individuals. Group A was older, and the experiment lasted a year, Group B was younger, and the experiment lasted five months. Because of the small groups, one can only speculate which factor (age vs. length of the follow-up) contributed more to the results. Coping in shift work is varied at different ages. In the study, the mean age difference between the groups was about seven years (14 years for the youngest and 4 years for the oldest subjects). This is quite marginally among working-age trained professionals. However, we tested this by analyzing the outcomes separately for duration and age group, and the results remained the same. Unfortunately, the small sample size and consequent small experimental groups did not allow for the formation of analyses of subgroups such as different age categories.

Group B subjects had, on average, more sleep loss and they were more tired at the beginning of the morning shift than subjects of Group A on average. The reason for this could be the short duration of the intervention and the adaptation to the new system is in a way still in progress. Anyway, the overall results of sleepiness for example, showed that the interaction effect of schedule change and group was not statistically significant, so it can be estimated that the age difference between the groups is not the major reason for the observed decrease in sleepiness in the new shift system. On the other hand, so-called healthy worker effect, i.e., the oldest represent the most adaptable in shift work. With caution, one could assume that our results could be interpreted as the new shift system benefited older workers.

In a survey, responses are always self-reported, subjective, and retrospective. The subjects involved were fully able to work and provided with comprehensive occupational health care, so help was available for any health problems that might arise. The workload was not studied, but the work can be considered mentally and physically moderate, but time pressure does occur. It may be reflected in the experience of work stress, which was generally at an acceptable level.

The subjective experience of the inconvenience of an early morning shift, a late evening shift, or a long night shift should, of course, be considered where possible. On the other hand, subjects had a good influence on the new shift schedule and did not return to the earlier shift system after the intervention ended. The developed shift schedule seems to be a rather good compromise for both employees and the employer when compared to successive morning and night shift periods, during which the accumulation of sleep debt is likely.

It seems that quickly forward rotating shift systems make it easier to adopt the circadian functions as shifts delay as indicated in longer sleep periods, and enhanced alertness at work. The new shift scheduling system also gives a better recovery by allowing enough time for recovery between individual shifts. Despite some of the changes lead to equivocal working hour characteristics i.e., long early morning or long night shifts, the new shift schedule showed positive effects on alertness at work, mitigated stress, and raised the satisfaction with shift system. In principle, the similar shift schedule could be functional also in other occupations working 24/7 with high task demands. 

## 5. Conclusions

The fast forward rotating shift schedule provides good sleep opportunities and assess alertness during work and free time. Better sleep and alertness also enhance work ability and reduce stress among aircraft inspectors in follow-up of the shift schedule change. The balance between the circadian factors (morning, evening, night shifts), the length of a work shift, and also the start and end times is important. Hence all these working hour characteristics in the rotating shift schedules besides the length of continuous free time should be accounted for while designing shift schedules.

## Figures and Tables

**Figure 1 ijerph-18-08105-f001:**
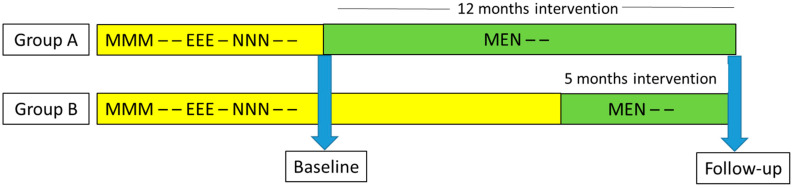
Study design. Earlier shift schedule in yellow: M = morning shifts 5.30–14.00, E = evening shifts 13.30–22.00, N = night shifts 21.30–6.00, – = free day. New shift schedule in green: M = morning shifts 5.30–15.30, E = evening shifts 12.00–22.00, N = night shifts 20.00–6.00, – = free day. Blue arrows indicate the survey times.

**Figure 2 ijerph-18-08105-f002:**
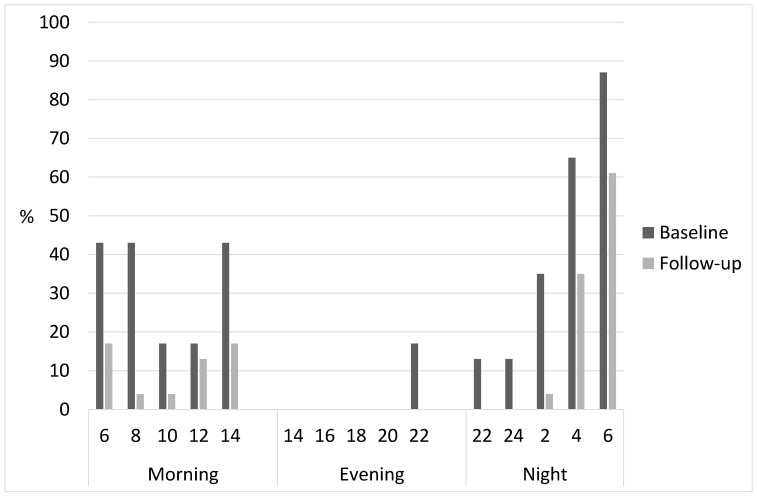
The shift-specific retrospective severe sleepiness (% of KSS ≥ 7) in every second hour of work shifts at the baseline and at the follow-up of the schedule change with combined Groups A and B (*n* = 23).

**Table 1 ijerph-18-08105-t001:** Characteristics (mean, SD, minimum, maximum) of the subjects in baseline, *n* = 23.

Intervention	Group A, *n* = 10 12-Months				Group B, *n* = 13 5-Months				*t*-Test
	Mean	SD	Min	Max	Mean	SD	Min	Max	*p*-Value
Age, years	50.7	4.1	45.0	58.0	43.5	6.3	31.0	54.0	0.005
Work experience, years	28.9	5.1	20.1	35.7	21.1	9.1	0.5	36.0	0.02
Current work, years	6.7	8.6	2.4	29.8	8.2	7.8	0.2	27.1	ns
Shift work, years	26.1	9.5	3.0	35.7	10.5	6.0	0.5	20.8	0.0001
Current shift work, years	4.6	5.1	1.5	18.8	4.1	5.4	0.0	20.8	ns

ns = statistically non-significant (*p* < 0.05).

**Table 2 ijerph-18-08105-t002:** Effects of schedule change and trial group on sleep (mean, SD) at the baseline and at the follow-up.

	A (*n* = 10)				B (*n* = 13)							
	Baseline		Follow-Up		Baseline		Follow-Up		Schedule		Group	
Dependent Variables	Mean	SD	Mean	SD	Mean	SD	Mean	SD	F	*p*-Value	F	*p*-Value
Sleep time in working days, h	6.6	1.2	7.8	2.0	5.8	0.9	6.9	2.6	8.9	0.007	1.5	ns
Sleep time in free days, h	8.1	0.9	9.4	1.4	8.0	0.7	9.2	2.1	16.9	0.0005	0.1	ns
Sleep need, h	7.9	1.2	7.8	2.1	8.2	0.7	8.8	1.3	0.6	ns	2.5	ns
Sleep loss, h	1.3	1.1	0.0	2.7	2.3	1.1	1.9	2.9	2.0	0.0006	5.0	0.04

ns = statistically non-significant (*p* < 0.05).

**Table 3 ijerph-18-08105-t003:** Effects of schedule change and trial group on the shift-specific retrospective alertness (KSS, average, SD) as a dependent variable during every second hour of work shifts at the baseline and at the follow-up.

	A (*n* = 10)				B (*n* = 13)							
Shift Time	Baseline		Follow-Up		Baseline		Follow-Up		Schedule		Group	
	Mean	SD	Mean	SD	Mean	SD	Mean	SD	F	*p*-Value	F	*p*-Value
Morning												
6.00	4.9	2.0	3.4	1.3	5.9	2.6	4.9	2.4	20.3	0.0002	2.1	ns
8.00	4.1	2.1	2.6	1.3	5.8	1.9	4.5	1.9	21.6	0.0001	6.6	0.02
10.00	3.6	1.6	2.6	1.3	5.7	1.9	4.2	1.9	12.1	0.002	8.2	0.009
12.00	4.0	1.8	3.3	1.6	5.4	2.1	4.3	2.1	7.9	0.01	2.5	ns
14.00	5.3	1.8	4.7	1.6	5.6	2.2	4.2	2.0	9.4	0.006	0.0	ns
Evening												
14.00	2.5	1.2	1.8	0.8	2.9	1.7	2.2	1.0	7.0	0.01	0.8	ns
16.00	2.7	1.1	1.8	0.8	3.0	1.7	2.2	0.9	8.9	0.007	0.7	ns
18.00	3.0	1.1	2.0	1.1	3.2	1.5	2.3	1.2	9.7	0.005	0.3	ns
20.00	3.8	1.5	2.7	1.2	3.7	1.3	2.5	1.4	17.4	0.0004	0.1	ns
22.00	4.7	1.9	3.9	1.1	4.2	1.6	3.4	1.3	4.2	ns	1.2	ns
Night												
22.00	4.6	1.5	3.1	0.7	4.3	1.4	2.6	1.3	27.9	<0.0001	0.7	ns
24.00	5.0	1.3	3.3	0.9	4.5	1.6	3.2	1.6	19.6	0.0002	0.3	ns
2.00	5.9	1.5	4.2	1.4	5.6	1.7	3.8	1.8	15.8	0.0007	0.4	ns
4.00	7.7	1.2	5.8	2.0	6.7	1.8	5.4	1.9	12.8	0.002	1.5	ns
6.00	8.0	0.7	6.8	1.2	7.5	1.3	6.4	1.7	15.9	0.0007	0.9	ns

ns = statistically non-significant (*p* < 0.05).

**Table 4 ijerph-18-08105-t004:** Effects of schedule change and trial group on wellbeing variables (average, SD) at the baseline and at the follow-up.

	A (*n* = 10)				B (*n* = 13)							
	Baseline		Follow-Up		Baseline		Follow-Up		Schedule		Group	
Dependent Variables	Mean	SD	Mean	SD	Mean	SD	Mean	SD	F	*p*-Value	F	*p*-Value
Satisfaction with schedule	4.5	0.5	1.7	0.7	4.5	0.7	1.7	0.6	231.4	<0.0001	0.0	ns
Stress	2.2	0.8	2.0	0.8	3.1	0.6	2.5	0.8	5.4	0.03	6.5	0.02
Work ability	8.0	1.6	8.6	0.8	7.4	1.3	8.6	1.0	14.9	0.0009	0.4	ns

ns = statistically non-significant (*p* < 0.05).

## Data Availability

Not applicable.

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
