# Peer review of "Longer Work Shifts, Faster Forward Rotation—More Sleep and More Alert in Aircraft Inspection"

_ijerph, 2021, doi:10.3390/ijerph18158105_

Round 1
Reviewer 1 Report
This manuscript is a meaningful topic about shift schedules and falls within the scope of the journal. The manuscript is also well-designed and immensely readable as follows.
1) This manuscript is a meaningful topic about the work of elderly workers.
2) In the introduction, the literature survey was adequately conducted.
3) The conclusion and discussion were also appropriately conducted based on the research results.
However, some revisions are required as follows.
1) The explanation of data variables should be systematically explained in terms of independent variables, dependent variables, and variable types using Table.
2) The number of questionnaire samples is not large. It is required to explain whether the distribution of the measurement variable follows a normal distribution so that it can be t-test or F-tested.
3) In the methods, a t-test was performed to investigate the group difference. However, the statistic is expressed as F in the result table. It is necessary to supplement the analysis method and the expression method of the Table.
4) The limitations of this study should be described.
Reviewer 2 Report
The authors submitted a manuscript which investigates the effect of a new schedule rotation in a particular job, namely aircraft inspectors.
The aim of the study was to compare sleep, alertness, and work ability between two different schedules. Two groups were then compared.
The reviewer has major concerns about the methods and results sections.
Methods:
Figure 1 is supposed to help the reader in understand the design of the experiment. It needs to be improved indicating graphically where 'intervention' is.
It is not clear why the authors decide to investigate two different intervention periods (12 and 5 months respectively). The authors must address this point since, at the moment, this element generates confusion in the reader.
Minor:
line 72-73: earlier rotation were 8.5 and not 8 hours long according to what is stated from line 76 on.
figure 1: correct base-line with baseline;
line 113-114: clarify the WAI score, is 1 or 10 the best score?
Results:
the entire section mainly relies on the interpretation of the three proposed tables with the written section that appear too short for an original article type. The written section must be improved.
What is more, tables are far to be intelligible. The reader, who not always has a deep statistical knowledge, need to understand what the parameter 'F' is and why it is used (please clarify in the methods too).
The main issue of concern is that all the tables show full data belonging to baseline and follow up and only partial data belonging to the differences between the group A and B.
As table 2 and 4 shows a significant statistical difference between group A and B only for sleep loss and stress the column 'group difference' may be neglected and discussed prior in the text.
On the other hand table 3 needs to be rethought. As 'group difference' is always statistically significant baseline and follow up data according to group must be furnished. Once this point is addressed severe sleepiness indicator (KSS>7) need to be statistically compared between baseline and follow up taking into account the important age and working career difference between group A and B. It is reasonable to believe that younger worker with shorter shift working career can cope better with sleep deprivation and may sustain all the apparent improvement in KSS>7 in morning and night schedule.
Discussion:
line 170-172 sentence is not clear and need to be rephrased. The author state that most part of the work of aircraft inspectors is concentrated overnight due to airline companies needs. Can this point sustain part of the higher stress, and the consequent sleep needed, highlighted in late night-early morning hours of the new schedule? is the evening rotation 'easier' because of that? the answer to these question should be included in the text.
The entire discussion section, again, does not refer to the group A and B and their difference or similarities, is this distinction really necessary?
The discussion section will need to be adapted to the changes required in the results section.
Limitations of the study are neglected. In the reviewer opinion major limitation is represented by the inhomogeneous workers characteristics between the group A and B and the questionary assessment just in two timepoints. Please include these and other limitations in a limitations paragraph.
Round 2
Reviewer 1 Report
Thank you very much for your new revised version of the manuscript. The revised manuscript is well-designed and quite readable. I think the article meets the standard of your journal.
Author Response
"Please see the attachment."
Reviewer 2 Report
The authors extensively remodel the critical manuscript sections according to the reviewer report. The reviewer appreciates it and believes that the manuscript is now more solid and, consequently, easier to read.
A few doubts still remains.
1) 'F' parameter issue has not been addressed in the methods section so its interpretation in the results appears still confusing;
2) A possible leading role of younger and, possibly, healthier group B in severe sleepiness (KSS>7) improvement in the new shift has not been discussed.
The reviewer appreciates the new structure of the discussion section spreading limitations among strength points.
Author Response
"Please see the attachment."
